# Predicting corporate credit risk: Network contagion via trade credit

**Claudia Berloco[1,2]\*, Gianmarco De Francisci Morales[3]\*, Daniele Frassineti[4], Greta Greco[1], Hashani Kumarasinghe[5], Marco Lamieri[1]\*, Emanuele Massaro[3], Arianna Miola[6], Shuyi Yang[1,2]**

**1** Intesa Sanpaolo, Torino, Italy, **2** Università degli Studi di Torino, Torino, Italy, **3** ISI Foundation, Torino, Italy, **4** Novartis Farmaceutica, Barcelona, Spain, **5** Zone24x7, Colombo, Sri Lanka, **6** Intesa Sanpaolo Innovation Center, Torino, Italy

\* claudia.berloco@intesasanpaolo.com (CB); gdfm@isi.it (GDFM); marco.lamieri@intesasanpaolo.com (ML)

## Abstract

Trade credit is a payment extension granted by a selling firm to its customer. Companies typically respond to late payments from their customers by delaying payments to suppliers, thus generating a ripple through the transaction network. Therefore, trade credit is as a potential vehicle of propagation of losses in case of default events. The goal of this work is to leverage information on the trade credit among connected firms to predict imminent defaults of firms. We use a unique dataset of client firms of a major Italian bank to investigate firm bankruptcy between October 2016 to March 2018. We develop a model to capture network spillover effects originating from the supply chain on the probability of default of each firm via a sequential approach: the output of a first model component on single firm features is used in a subsequent model which captures network spillovers. While the first component is the standard econometrics way to predict such dynamics, the network module represents an innovative way to look into the effect of trade credit on default probability. This module looks at the transaction network of the firm, as inferred from the payments transiting via the bank, in order to identify the trade partners of the firm. By using several features extracted from the network of transactions, this model is able to predict a large fraction of the defaults, thus showing the value hidden in the network information. Finally, we merge firm and network features with a machine learning model to create a 'hybrid' model, which improves the recall for the task by almost 20 percentage points over the baseline.

## Introduction

The goal of the paper is to shed light on the main determinants of firm distress by focusing on the trade credit channel as a key source of contagion, thus improving the forecasts of the default probabilities of firms in the short-term. We design an 'hybrid' network-based early warning system that combines statistical tools with machine learning techniques, leverages a huge dataset of financial transactions, and aims to maximize the out-of sample predictive performance.

**Data Availability Statement:** The data that support the findings of this study are available from Intesa Sanpaolo but restrictions apply to the availability of these data, which were used under license for the

current study, and so are not publicly available. Data are available upon request from the following contacts: - Intesa Sanpaolo Innovation Center <innovationcenter@pec.intesasanpaolo.com> - Intesa Sanpaolo Data Office <dc_do_dat_sci_art_in.14388@intesasanpaolo.com> The authors did not receive any special privileges in accessing the data. Given the sensitive nature of the data, the authors whose affiliation is not Intesa Sanpaolo had to sign a non-disclosure agreement (NDA) to get access.

**Funding:** Arianna Miola is employed at Intesa Sanpaolo Innovation Center; Claudia Berloco, Greta Greco, Marco Lamieri, Shuyi Yang are employed at Intesa Sanpaolo. The funder provided support in the form of salaries for these authors (AM, CB, GG, ML, SY), but did not have any additional role in the study design, data collection and analysis, decision to publish, or preparation of the manuscript. The specific roles of these authors are articulated in the 'author contributions' section.

**Competing interests:** The authors of this manuscript have read the journal's policy and have the following competing interests: Arianna Miola is employed at Intesa Sanpaolo Innovation Center; Claudia Berloco, Greta Greco, Marco Lamieri, Shuyi Yang are employed at Intesa Sanpaolo; however, this does not alter our adherence to PLOS ONE policies on sharing data and materials.

Most of the bankruptcy models, as described in Section Model-based approaches, focus on the idiosyncratic characteristics of firms and on market conditions, while we direct our attention towards the interconnections among firms, by leveraging the complex network of interdependent firms linked by financial exchanges.

Firms are not independent agents, rather, they exchange information, goods, and services in order to function and enable the production of value. Supply chains enable flow of materials, products, and associated information downstream, while cash flows upstream towards manufacturers and producers of raw materials. These flows link customers and suppliers in a network of financial transactions.

In this context, trade credit is a payment extension granted by the suppliers to their customers. The interdependency of firms in trade credit networks allows shocks and imbalances to propagate along the supply chain, and in the entire economic system. A shock to the liquidity of some firms may cause a chain reaction in which other firms also suffer distress. When liquidity tensions arise, firms are likely to default on trade credit payments (i.e., to accumulate trade debt), especially the financially constrained ones.

The use of trade credit by firms is controversial. A large strand of the literature posits that trade credit is an important source of financing for manufacturing firms, especially for those firms which experience temporary distress and are more vulnerable to financing constraints, such as small and medium-sized enterprises (SMEs) [1–4]. In fact, suppliers of trade credit own an implicit stake in the customers' business: they own strong incentives to provide credit to financially distressed clients, or to clients that experience liquidity shocks, to maintain a product-market relationship and to preserve their future earnings.

Among the most recent contributions on the buffer-stock role of trade credit, McGuinness et al. [5] examine if trade credit helped financially constrained SMEs survive the recent crisis. By using data across thirteen European countries over the period 2003-12, they show that trade credit had a large positive impact on firm survival. Moreover, they report evidence of a significant redistribution effect, with cash rich or unconstrained firms extending significantly more trade credit than their less financially resourced counterparts. Similar studies [6–14] support the same view that trade credit provides a useful buffer for financially constrained firms. In general, small firms and credit-constrained firms are likely to rely more on supplied trade credit, especially during contractionary phases, when bank credit is rationed [15, 16].

Most importantly, trade credit can be regarded as a potential vehicle of propagation of losses in case of default events. Trade credit interconnections might act as amplifiers of firm shocks. In a network of firms that borrow from each other, a temporary shock to the liquidity of some firms may cause a chain reaction in which other firms also suffer distress: firms respond to late payment from customers by delaying payments to their suppliers. This holds particularly true during a recessionary phase, when a global lengthening of payment terms occurs. Contagion effects may in turn translate into a large and persistent decline in the aggregate activity.

Kiyotaki and Moore [17] provide a seminal contribution on trade credit chains as the channel through which liquidity shocks are propagated in the economy. Based on their ideas, Cardoso-Lecourtois [18] develops a general equilibrium model that deals with aggregate consequences that a temporary (productivity or liquidity) shock might have on the whole economy. Results show that in a carefully calibrated model, the effects of these credit chains are quite important. In addition, Raddatz [19] proves empirically that trade credit chains are at the origin of higher output correlation between industries. More generally, Gabaix [20] analyzes how the propagation of idiosyncratic shocks through inter-firm linkages has macroeconomic relevance.

The aggregate effects of trade credit chains have been highlighted in the financial network literature as well, arguing that counterparty exposures may cause shock propagation and systemic failure, mainly through input-output linkages [21–23]. In a networked economy, the failure of one firm may have a snowball effect, causing failure of other firms, and in extreme cases causing an avalanche of failures, known as a bankruptcy chain [22, 24]. Hurd [25] provides an exposition on mathematical stochastic models of contagion and cascades in the context of systemic risk, and recent work proposes models that are specific to the case of credit risk [26]. Chain-reaction bankruptcies, a bankruptcy triggered by a proceeding default of a firm that trades or has other relations with the bankrupting firm, have also been empirically studied recently [27, 28].

With the new availability of microdata, contagion effects in trade credit chains have started to be explored in a greater detail, at least providing survey or indirect evidence on the importance of the propagation mechanism [24, 29–35]. Specifically, Boissay and Gropp [34] exploit a unique dataset on trade credit defaults among French firms to show that entities which face idiosyncratic liquidity shocks are likely to default on trade credit payments: a negative liquidity shock is shown to be transmitted along the trade credit chain, at least until it reaches a trade creditor with access to external financing, or sufficient cash-holdings. Similarly, Jacobson and von Schedvin [24] use a Swedish data set on claims held by trade creditors (suppliers) on failed debtors (customers) to quantify the importance of trade credit chains for the propagation of corporate bankruptcy. They show that suppliers of credit who are exposed to credit losses due to failing customers are in turn subject to an elevated risk of failure. It is also worth mentioning a recent development of their work, jointly with Amberg et al. [36]. The paper focuses on how firms manage liquidity shortfalls by using their cash reserves and increasing their trade credit margins. The authors show that adjustments in the amount of trade credit drawn from suppliers are more pronounced compared to the adjustments in the amount of trade credit issued to customers. This finding suggests once again that the dominating contagion effect goes upstream (from debtors to creditors).

Lamieri and Sangalli [37] investigate the relationship between trade debt and firm solvency with spatial econometric techniques. They focus on the contagion effects which originate from the manufacturing supply chain exploiting a network of inter-firm transactions executed before the outbreak of the 2008 crisis. They propose a two-step econometric framework that deals with contagion effects in trade credit chains. The first step is a SAR spatial model that accounts for spatial lag dependence in stocks of trade debt pertaining to interconnected firms, i.e., spillover effects of trade debt. In the second step, trade credit chains are investigated as a key determinant of the firm's likelihood of distress, jointly with variables that proxy for the bank-firm relationship. Estimation results show that trade debt has been affected by the liquidity and financial status of a firm and by spatial neighborhood effects. A positive spatial autoregressive coefficient in the first step of the model can be interpreted in favor of a chain reaction at work during the crisis, following a progressive lengthening of payment terms which simultaneously affected the interconnected firms in our manufacturing supply chain. The phenomenon is found to exert a positive impact on distress likelihoods. The estimated effect of fitted trade debt is pretty much comparable to the impact of financial rigidity, measured by firm leverage, which is well established in the literature on corporate distress. This result is crucial, and sheds light over the need to account for complex interactions between firms in the analysis of the solvency behavior, at both the individual and systemic levels.

The evidence summarized so far points towards the existence of a phenomenon of imbalance propagation via inter-firm links. If this fact is true, all the models that are designed to evaluate the solvency behavior of firms should take these linkages in a more careful

consideration. For this reason, in this paper we explore how to integrate network effects due to trade credit into forecasting models for the imminent default of a firm.

We propose a hybrid classification model for forecasting a firm's default. A first module which uses only features of the economic trends of the single firms. The output of this module is used in a second one in which only network features are used. This second module looks at the transaction network of the firm in order to identify its trade partners. While the first module is the standard econometrics way to predict such dynamics, the network module represents an innovative way to look into the firms' fragility to shocks originated from connected firms, and propagated through the trade credit channel. The model which uses only network information is able to predict a large fraction of the defaults, thus showing the value hidden in the network information. We then merge firm and network features by using a machine learning model, thus creating a 'hybrid' forecasting model. The features are selected via a systematic data-driven process, guided by human evaluation. The final hybrid model is able to substantially improve over a single-firm-only baseline on all measures of interest, such as Recall, Precision, and AUC.

## Materials and methods

### Basic definitions

Here we report some relevant concept and definitions.

**Definition of default.** We wish to predict which firms will experience financial distress, and are therefore likely to default on their loans towards the banks.

The standard definition of a default event is, according to regulations (the definition of default was introduced by Directive 2006/48/EC of 14 June 2006 (Capital Requirements Directive—CRD), later replaced by Regulation (EU) No 575/2013 (CRR). The definition of default of an obligor specified in Article 178 of the CRR includes, inter alia, the days past due criterion for default identification, indications of unlikeliness to pay, conditions for a return to non-defaulted status and treatment of the definition of default in external data), when a firm is past due a loan payment for 90 days or more, given the application of a relative materiality threshold of 5% (following the financial crisis, the European Banking Authority (EBA) has established tighter standards around the definition of default to achieve greater alignment across banks and jurisdictions. A new definition of default need to be implemented by banks by the end of 2020).

Here, we use a broader definition of "integrated default", that includes firms which are defaulted and also firms who have a low credit rating as per Advanced Internal Rating-Based (AIRB) model, and are therefore unlikely to repay their loans in case of financial shocks or liquidity shortage. Henceforth, we refer to this notion of "integrated default" simply as default.

**Target variable.** We assume a monthly time series of binary indicators $D$, indexed by time $t$ and firm $i$, such that $D_i^t = 1$ if and only if firm $i$ is in default at time $t$. We use a time window for the forecast of the default of 3 months, which represents a typical short-term horizon for financial distress (one quarter), aligned with typical early warning models [38].

That is, given a firm at time $t$ currently in bonis (i.e., not in default $D_i^t = 0$), we wish to predict whether the firm will be in default at any point in the interval $[t + 1, t + 3]$. Thus, our target variable $Y_i^t$ is a logical 'or' of lagged versions of $D_i^t$ up to 3-months:

$$Y_i^t = D_i^{t+1} \vee D_i^{t+2} \vee D_i^{t+3}.$$

## Data description

Data is drawn from a proprietary dataset belonging to Intesa Sanpaolo (ISP), the leading Italian commercial bank. The dataset is highly representative of the sectoral structure of the Italian economy, with a prevalence of the manufacturing industries (see statistics from http://dati.istat.it/, section 'Imprese'. The main sectors by value added are: mechanical engineering, automotive, metallurgical, food and beverage).

Our analysis is based on two main types of datasets. The first one is a large panel dataset of corporate data, with a monthly frequency. The most important set of informations for each firm include:

- Structural information about the firms:

  - micro-sector of economic activity, classified according to 5-digits NACE codes (https://ec.europa.eu/eurostat/statistics-explained/index.php?title=NACE_background);

  - geographical location of the headquarter: town, province, and region according to the NUTS classification (https://ec.europa.eu/eurostat/web/nuts/background);

- Yearly financial statements: balance sheet and income statements reclassified according to the CEBI principles (CEBI is the Central Financial Statements area of Cerved Group that developed a platform for the management of economic and financial information in terms of acquisition, storage, control, reclassification and evaluation. See https://www.cerved.com/en/offering/banks-finance/credit-risk/customer-assessment). The list of the reclassified financial statements items is available in the S1 Appendix;

- Bank of Italy Central Credit Register (the Central Credit Register (CR), managed by the Bank of Italy, is a database on firms' debts towards the banking system. The CR is supplied with data that the participating intermediaries (banks, financial companies, and other intermediaries) send in relation to loans and guarantees granted to their customers, to guarantees received from their customers, and to loans or guarantees purchased from other intermediaries. Intermediaries classify a customer as a bad debtor and report them as such to the CR when they believe that they are in serious difficulty with loan repayments. The Bank of Italy informs participating intermediaries about the overall indebtedness of their customers, the types of loans they have had, and whether they have made their payments. See https://www.bancaditalia.it/statistiche/raccolta-dati/centrale-rischi/index.html) data on the overall indebtedness of customers by types of loan, the main variable being customers' total granted loans amounts and customers' total used loans amounts, either short-term or long-term (the dataset full description containing all the variables specification is available in the Bank of Italy memo num. 139, available at https://www.bancaditalia.it/compiti/vigilanza/normativa/archivio-norme/circolari/c139);

- Overdraft (an overdraft is an extension of credit from the bank that is granted when an account reaches zero balance. The overdraft allows the account holder to continue withdrawing money even when the account has insufficient funds to cover the amount of the withdrawal) loans amount and number of days since the customer is not paying the installment;

- Customer's regulatory risk parameters computed by the bank: Probability of Default (PD), Loss Given Default (LGD), Exposure at Default (EAD), Expected Credit Loss (ECL), and Risk Weighted Assets (RWA). Namely, the Advanced-IRB approach of the Basel Capital Accord provides a single framework by which a given set of risk components or inputs are translated into minimum capital requirements. Banks can use this approach only subject to

approval from their local regulators (for more details see https://www.bis.org/basel_framework);

- Credit risk alerts issued by the bank's current Early Warning System: the current model clusters the portfolio into 4 classes according to their short-term (3 months) probability of default of the firm, using only features pertaining to the firm, thus ignoring firm-to-firm connections and interdependences.

Every database is reduced to a monthly frequency either by replicating a static/yearly information or by summarizing daily data into monthly measures. The sample includes more than 600 000 firms that are customers of the bank and their data is observed in the period from Q3 2016 to Q1 2018.

Out of this sample, we are interested in predicting the default of a subset of important firms, which are relevant from the business point of view for the bank. These firms are those classified in regulatory terms as Corporate and SME Corporate, thus excluding small firms, with turnover lower than 2.5 million Euro, and personal companies. We call this set of firms the target perimeter $P_T$. However, the overall data refers to a larger set of firms, which are nevertheless useful to model in view of the risk contagion process. We call this larger set of firms the extended perimeter $P_E$.

The $P_T$ perimeter represents a small fraction of the entire dataset, i.e., 16%. In Fig 1 we report the difference between the number of firms in the *target perimeter* and the ones in the extended one. Every month we have information about 648 893 active firms on average, of which 107 103 are firms in $P_T$.

The second type of data is the network which represents the supplier-customer relationships. The network is built by leveraging both cash payments and delayed payments between firms. Among cash payments we collected data on Bank Transfer, Invoice Payments ("Ricevute Bancarie'" in Italian) and SEPA (Single Euro Payments Area) Direct Debit. Among delayed payments we consider both Invoice Discounting Facilities and Factoring. In this network we represent firms as nodes and payments as edges. The details on how the network is defined are discussed in the following section, but the order of magnitude of the network is 600 000 nodes

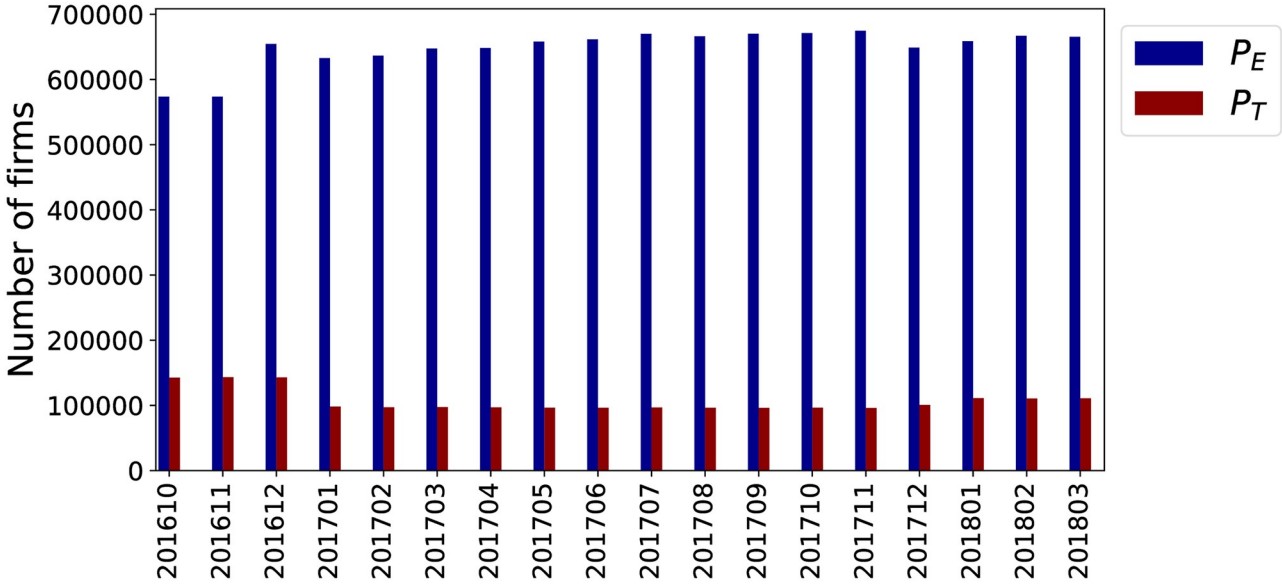

**Fig 1. Size of the target perimeter $P_T$ and the extended perimeter $P_E$ across time.**

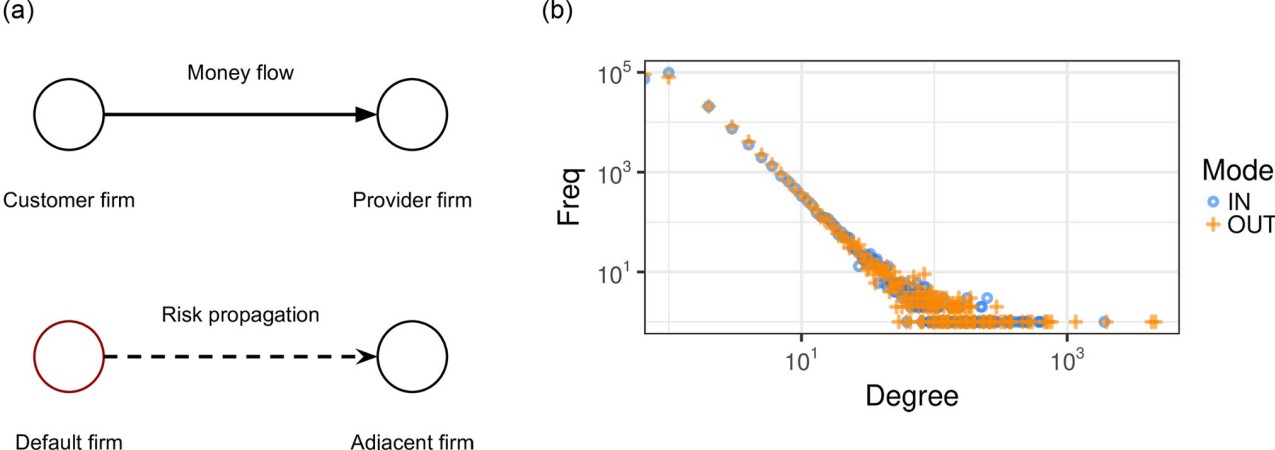

**Fig 2.** (a) A component of the inter-firm business transaction networks. The nodes are firms and links are business transactions from customers to providers, indicating a money flow and thus a possible trade debt. In a similar way, we assume that also the risk flow, i.e., propagates, between adjacent nodes. The nodes are firms and links are risk propagations from an insolvent firm to its adjacent nodes. (b) degree distribution of the transactions networks (blue points correspond to incoming degree ($k_{in}$) while orange to outgoing degree ($k_{out}$)). The degree of the networks shows a typical scale free distribution, i.e. $y \sim k^{\gamma}$: here $\gamma = -2.54$ and $\gamma = -2.50$ for the incoming and outgoing nodes' degree distribution.

and 4 million edges, which represent an average of 23 million yearly transactions along the 18 months. Our main working assumption by using this data is that the transaction network is a proxy for the trade credit network. This assumption is supported by theoretical and empirical evidence [39], especially in the Italian case, where trade credit is extremely common [40].

**Network of transactions.** Our dataset contains lists of business partners for each firm. We can draw a directed edge for each pair of business partners, which indicates the money flow, as shown in Fig 2(a). The opposite direction of the edge is the direction of flow for goods and services. The data contains transactions from firms which are clients of Intesa Sanpaolo (ISP) from 2016 to 2018. For each month we aggregate the information of transactions between the firms in the past 12 months. We explored different temporal aggregations and observed that different lengths for the aggregation window affect neither the structure of networks nor the dynamics of the spatial diffusion. A temporal analysis of the networks is reported in the S1 Appendix.

We model the interaction among firms as a weighted directed graph $G(V, E, w)$, where $V$ is the set of firms, and $E$ is the set of directed edges $(u, v)$ which represent that a firm $u$ is a customer of the firm $v$, and therefore its credit risk can spread along the direction of the edge. The weight of the edge $w(u, v)$ represents the strength of the connection between the two firms, and is computed as the total value of the transactions within the aggregation window. In addition, the contagion is a dynamic process that happens on the network throughout time. Thus, we need to model both dimensions (i.e., space –the network– and time) at the same time in order to capture the characteristic of the process. Indeed, the graph itself changes in time, as the relationships among the firms evolve. Therefore, we model our data as a dynamic graph $G^t = (V, E^t, w^t)$, where the set of vertices is fixed, but the edges change with time. The analyzed network obtained by projecting the dynamic graph along the time dimension shows a typical scale free distribution of the connectivity degree as shown in Fig 2(b). Similar complex scaling relations have been observed also in a similar study of Japanese inter-firm network data [41].

The complex scaling of the inter-firm networks is also observed in the distributions of the number and the total amount of transactions. Also in this case we observe in Fig 3(a) and 3(b) a heavy-tailed distribution as well as a symmetric pattern between customers and suppliers.

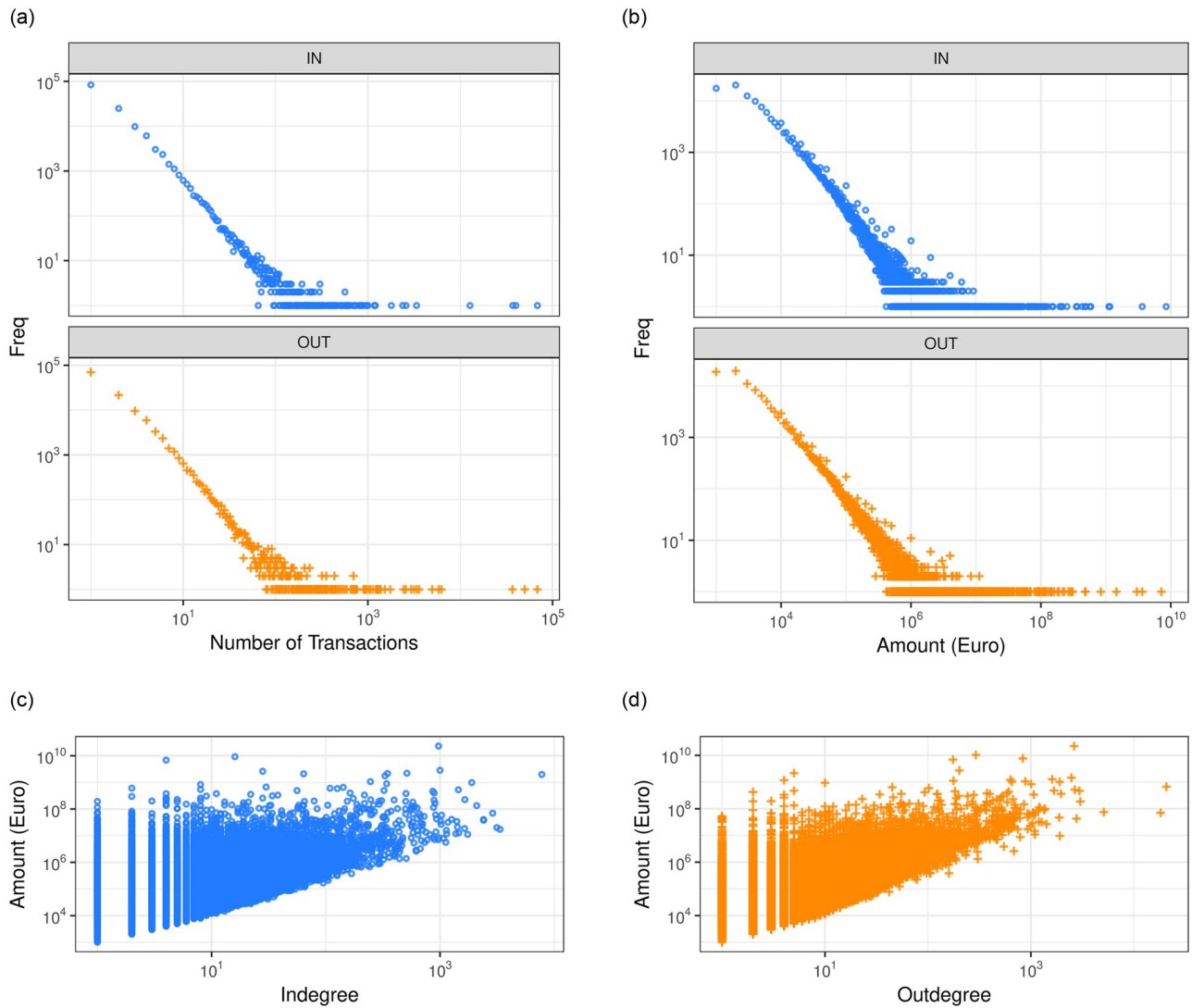

**Fig 3.** Distribution of (a) the number and (b) the amount of transactions. (c-d) Scaling relations between amount of transactions and degrees in log-log scale.

Fig 3(c) and 3(d) shows the scatter plots for amount of the transaction and node degrees in logarithmic scale. We also find a superlinear scaling relation, $y \sim x^\gamma, \gamma \sim 1.3$ for both in-degree and out-degree considering the amounts. Clearly, the network structure of firms plays an important role in determining the sales, as already observed by Tamura et al. [41]. This fact implies that larger amount of money exchanged is correlated with a larger number of connections.

We now wish to analyze the portion of the client annual revenue (inferred from the balance sheet) that is received on ISP accounts from business partners also sending money from an ISP account, considering the fact that both the parties are likely to rely on multiple-bank accounts. This fraction of revenue corresponds to how much visibility the bank has into the trades of each firm.

**Transactions / revenues analysis.** Fig 4 shows the distribution of this quantity for firms in the target perimeter $P_T$. The average knowledge of the transactions of the ISP firms is

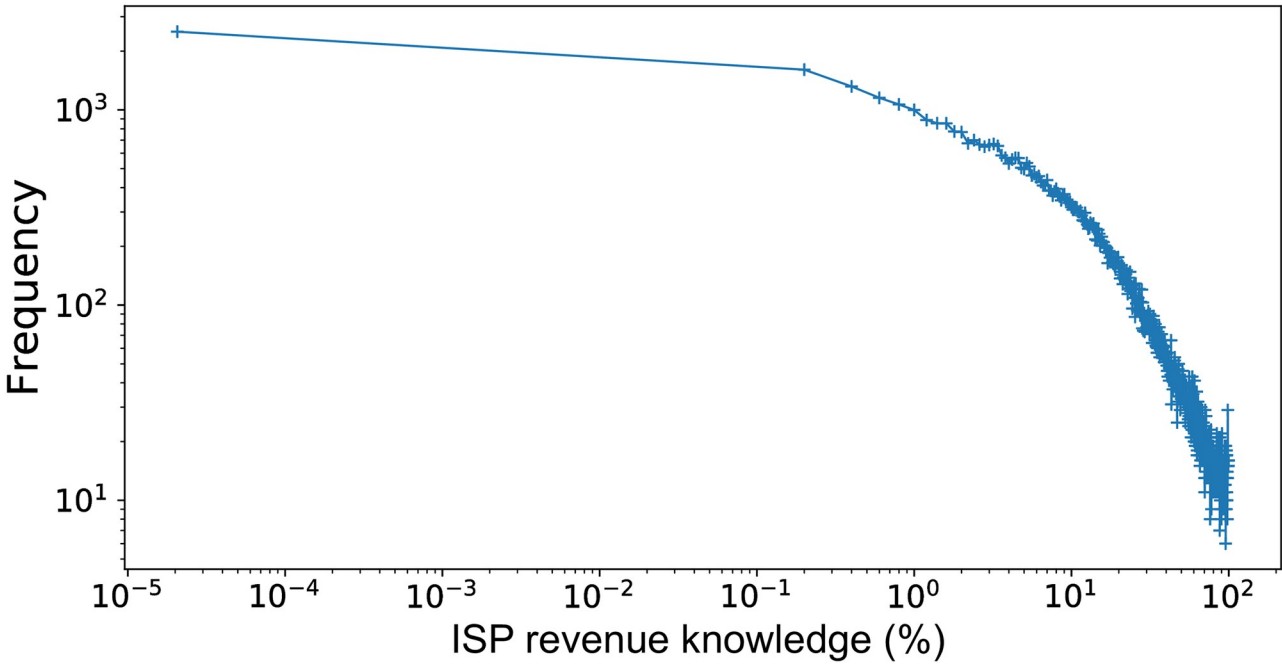

**Fig 4. Distribution of the relative knowledge of the volume of transactions in comparison with the total revenue for the target perimeter.**

around the 16%, and the median value is 9%. The dataset has information about 90% of the volume of transactions for only 1% of the firms, while at the 95-percentile of the firms only 0.25% of the revenue coverage is available. Indeed, the distribution in Fig 4 appears quite skewed. Next, we show how to increase this coverage by carefully applying a machine learning model for record linkage.

**Network enrichment.** Although multiple-bank relationship is widely spread abroad, in Italy this phenomenon is four times more significant than the median in Europe [42]. In addition, the number of banking relationship generally increases as the size of the firm grows. Data from the Credit Registry shows that 16.6% of the firms have relationship with up to three banks, what is surprising is that one in four firms deal with more than 10 banks. This framework reflected in our dataset: over 94% of the transaction are carried out between an ISP customer and an external unknown counterpart, of which all we know is an IBAN (international bank account number) and a description of the name.

If we consider a single transaction (e.g., a bank transfer) we are able to draw a link between the business partners only when both the sender and the receiver exchange money on an ISP bank account. It follows that, if one of the parties carries out the payment using an account held with another bank, the information is not available in our network. Accordingly, initially, we are able to trace on average the origin of 16% of annual revenues for clients in the target perimeter.

In order to increase the fraction of firm annual revenue captured by our network, we set up a *network enrichment* task. Considering a bank transfer transaction involving an external client identified by an IBAN and a firm name (an arbitrary string), we wish to check whether such unknown firm has an account in ISP. Namely, we want to determine whether it was possible to find a reasonable match between the firm name linked to the external IBAN and a firm name in the ISP database. This task is commonly known as *record linkage* [43].

**Table 1. Performance of the model for record linkage on the test set.**

| | |
|---|---|
| Precision | 99.98% |
| Recall | 73.03% |
| F1 measure | 84.45% |

In bank transfers, the firm name is a free-text field, susceptible to mistakes and typos, and it is often followed by the ownership structure which may appear in abbreviated form (e.g., "srl', "s.r.l.", or "società a responsabilità limitata", Italian for limited liability company). The legal structure may comprise a number of characters larger than the firm name itself. Due to the wide variety of different ways to refer to the same firm, even inside the database of ISP, standard string comparison methods and unsupervised models are not likely to perform adequately.

However, the variability of the spellings of the same firm inside the ISP firm registry provides valuable data to train a supervised model. Each firm in the registry has a unique identifier, and several possible spellings of the firm name. We generate positive training examples by picking all pairs names referring to the same firm. Correspondingly, we create a set of random pairs of firm names to create negative examples. We use regular expressions to standardize the appearance of uneven legal structures, and transform them into acronyms so that they represent just a small fraction of the total string length.

For each pair, we compute standard string distance metrics to use as features, such as edit distance, cosine distance, Jaccard distance, and q-gram distance. We train a logistic regression model on this data, which turns out to have an extremely high precision, at the expense of the recall, as shown in Table 1. This result suits our purposes since we prefer being conservative and discard a real connection rather than inferring a non-existent one.

Once the model is trained, we still need to devise an application strategy, as trying every possible pair of firms in a database with several hundred thousands firms is infeasible. Here, we leverage again the multiple-bank phenomenon: if a client firm holds account with different banks, then it is likely to transfer liquidity from a financial institute to another. Therefore, we only test pairs of firms that are linked by a bank transfer. Thus, we are able to identify the external IBANs of client firms that transfer money between ISP and another bank, and we use this match to identify the same firm (via its IBAN) as it makes transactions with other ISP customers.

As a result, considering the monthly network data, we increase the amount of traced transactions by 450%, the coverage by 200%, and get from 281.203 links to 826.688 links in a month.

**Neighbourhood analysis.** Here we explore the idea that firms in default are part of a risk contagion process by looking at the default probability of the nodes in the networks (a more thorough analysis of this idea is presented in the S1 Appendix). Fig 5 shows the probability of default $P(d)$ for nodes who made transactions with nodes that went in default in the past for different aggregation windows 3, 6, 9, 12 months. $P(d|k(d))$ is computed in the following way: for each node that is in default at time $t$ we save the adjacent nodes and we count the fraction of them that went in default within time $t' = t + \delta$, where $\delta \in \{3, 6, 9, 12\}$ months. Clearly, the higher the $\delta$ the higher value of $P(d|k(d))$ we can expect. However, this result suggests that the direct contagion effect is on average low. Indeed, a firm can transmit risk even without going into default itself, for example by tightening the amount of trade credit it extends downstream in periods of distress. Therefore, If we want to leverage the trade credit to predict firms' defaults, we need a more nuanced approach.

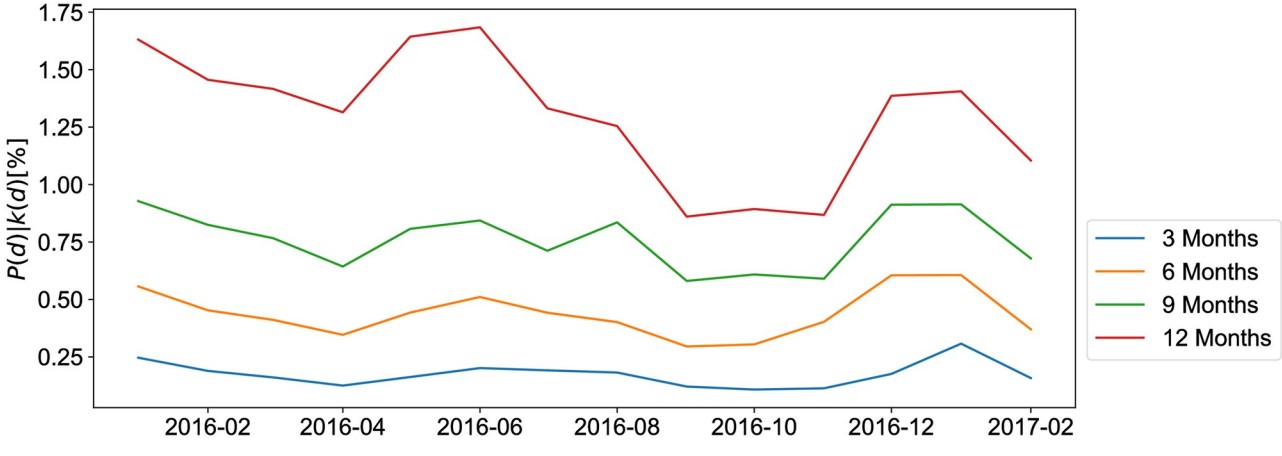

**Fig 5. Percentage of adjacent nodes of insolvent firms at time $t$ that experienced a default $\delta$ months later (for $\delta \in \{3, 6, 9, 12\}$).**

## Model-based approaches

Starting with the seminal contribution of Altman [44], a large number of corporate bankruptcy prediction models were developed [45]. They can be classified in two broad categories: structural models and reduced-form models.

Structural models [46], pioneered by Black, Scholes, and Merton, ingeniously employ modern option pricing theory in corporate debt valuation. The Merton model [47] was the first structural model and has served as the cornerstone. Structural models focus on the stochastic process of a corporate obligor's assets and postulate that a default occurs when firms' value cross a threshold value. The models can be divided into an endogenous default and an exogenous default groups. The frameworks in the former group let borrowers choose strategically the timing of default. In contrast, the models in the latter group impose an ad hoc default trigger but develop richer stochastic structures that capture empirical regularities of credit markets [48]. In the context of two endogenous default models, the de-facto benchmark for practitioners and banks is the commercial model of Kealhofer, McQuown, and Vasicek (KMV) developed by Moody's [49]. The KMV model measures the Expected Default Frequency (EDF), a forward-looking measure that is scaled to actual probabilities of default using a historical default database. The dynamics of EDF comes mostly from the dynamics of the equity values and its volatility. Therefore this approach is best suited for public listed companies, as in the US market. On the contrary, in the Italian market the vast majority of firms is private and not-listed, thus EDF can be calculated only by using some comparability analysis based on accounting data.

The structural approach, led by Merton's model, has the highly appealing feature of connecting credit risk to underlying structural variables. It provides both an intuitive economic interpretation and an endogenous explanation of credit defaults. The main disadvantage of structural models lies in the difficulty of implementation, the not always realist assumptions, and the difficult calibration of stochastic asset processes by using publicly available information. Furthermore, they tend to be analytically complex and computationally intensive [46].

Conversely, models belonging to the second category see bankruptcy as an exogenous events driven by a stochastic process and thus model it as a binary classification problem, by leveraging financial and non-financial data. Therefore, reduced-form models do not consider endogenous cause of defaults; rather, they rely on exogenous specifications for credit default and debt recovery. This feature is both a strength and a weakness—while these models suffer

from the lack of economic insights about default occurrence, they offer more degrees of freedom in functional form selection. Such flexibility contributes to analytical tractability, and ease of implementation and calibration. Most actuarial models used for credit risk measurement and probability of default estimations by Italian banks lie within the reduced-form class. Common statistical techniques [50] include linear discriminant analysis, multi-discriminant analysis, and logistic regressions. Machine learning techniques [51] includes artificial neural networks, Support Vector Machines [52], Bayesian network, and decision trees [53]. In addition, to improve the prediction, accuracy optimization algorithms, such as genetic algorithm [54] and particle swarm optimization (PSO), have been combined with machine learning techniques [55].

An Early Warning System (EWS) is a system that alerts for the probability of default before they affect the financial statements of firms. EWSs are used for detecting potential bankruptcies in the short-term, typically within a 3-months horizon. In essence, an early warning system is a reduced-form, short-term bankruptcy model used for managerial purposes, compared to the regulatory compliance purpose of an A-IRB rating models. The probability of default obtained by an EWS is not used to derive regulatory measures such as Expected Credit Loss (ECL) or Risk Weighted Assets (RWA), so it does not need to comply to all the requirements of an A-IRB rating model. The literature on EWS is more the domain of practitioners than academics. Given the higher degree of freedom in the model design, most of the recent EWS proposed in the literature [56, 57] are based on machine learning techniques.

The flexibility of reduced-form models and the more data-driven methodology makes this approach more appealing to tackle our research question, and to better leverage the huge dataset of financial transactions at our disposal. Moreover, our model design focuses toward a short-term (3-months) bankruptcy forecasting model that can be classified within the broad class of EWSs, in order to benefit from the lower regulatory requirements and higher freedom in model design.

We design an EWS by combining statistical tools with machine learning techniques, therefore using an hybrid modeling approach. The statistical part of the model, described in section Modeling fragility of firms to network spillovers, is more grounded in economic theory, while the machine learning component, described in section Hybrid model aims at maximizing the out-of sample predictive performance. We now present the two components of the proposed modeling approach in more details.

The first part is a network spillover model that stresses the fragility of the firm to shocks originated from the network to predict the target variable. The first step uses only the features of the single firms, while the second step uses network features along with the predictions of the first step. This model is inspired by spatial econometrics [37], and its results are easily interpretable since a Generalized Linear Model (GLM) is used for the second step. The goal of this model is to show the predictive power of the network information when it is not used in explicit combination with single firm features, which are already known to be significant in this context.

The second component consists in a machine-learning model, where firm and network features are used together to predict the target variable in a single step. This model aims at achieving a better predictive performance by leveraging the network information with respect to a single-firm baseline, after the importance of the network information was clarified by the exploratory analysis and the network spillover model.

In this hybrid model approach, the features' creation and selection are performed in a data-driven fashion, but supervised by a domain expert. In this case, the estimation and the predictions are less interpretable with respect to the network spillover approach, but a feature

importance analysis highlights how the network information integrates well with the usual single-firm features.

**Task definition and evaluation.**    One of the main applications of identifying risky business is to enact proactive measures to avoid the default from happening. However, the resources for these measures are limited, both in financial terms and in time needed to enact them. Therefore, the default forecasting task naturally lends itself to be modeled as a ranking task, where each firm is given a probability of default $P(d)$ and then the top-k firms in the ranking are flagged as risky. We naturally lean towards a probabilistic classifier, such as logistic regression, which models the probability of each instance to belong to the class of interest (default). Consequently, and given also the high imbalance of the class labels, our main performance metric is Recall@K. We choose K to be the top 5% of the perimeter, which is a realistic figure given the experience of ISP. Henceforth, we use R@K to refer to the Recall@5% of the firms under test (similarly, P@K indicates the Precision@5%).

**Prequential validation.**    Given the temporal nature of the data, it is natural to model the data as a stream, and thus apply stream mining or time series forecasting evaluation techniques (e.g., see Section 5.9 from [58]). In particular, we employ a *prequential validation* setting. We replay the data in temporal order as a stream, with each month representing a time step (the order of the firms within the month is not relevant). At time $t$, we first predict the label (or the $P(d)$) for each of the firms for which the features are available. Then, we reveal the labels to the model for training, and update it before moving to time $t+1$.

This prequential validation setting has several advantages:

- It simulates the operating conditions in which the model will be deployed in production.

- The performance numbers are always based on out-of-sample predictions, and are therefore immune to overfitting.

- All the available data can be used for training, rather than keeping some as holdout.

- The model is always up-to-date with the latest available data.

In the specific context, a sequence of 12 rolling training and test sets are created in order to cover possible seasonal effects: the training set with predictors available up to time $t$ for default events in $[t+1, t+3]$ and the test set predictors at time $t+3$ for default events in $[t+4, t+6]$. It is important to point out that taking features at time $t$ does not mean that the information set available for prediction is only the one of time $t$, in fact, during the features engineering process (see Section Feature Engineering) several variables are constructed to capture past months dynamics (trends and variation) of primary variables.

## Modeling fragility of firms to network spillovers

The goal of this module is to capture network spillover effects originating from the supply chain on the $P(d)$ of each firm.

Therefore, we employ a sequential modeling approach: the output of a first model on single firm's features is used in a subsequent network model. The former model captures the effect on the $P(d)$ of the single-firm's features, and predicts the $P(d)$ of each firm in isolation. The latter model captures the network spillovers, by leveraging the output of the first model, together with the features capturing the network structure and the position of the firm in the supply chain, to determine the influence of the neighborhood of each firm onto the $P(d)$ of the firm itself.

Given that we wish to use the $P(d)$ estimated by the first modeling step as input for the network spillover model, we define the perimeter of the first step (i.e., the firms to include in the dataset) to be as wide as possible, to preserve the network structure.

The first extended perimeter $P_E$ therefore includes all the firms available in the ISP database. The decision to base the network spillover effects estimation on the extended perimeter $P_E$ is linked to the difficulty of reducing a complex network without loosing some of its main features.

The perimeter for the network spillover model, instead, includes only the firms which we are actually interested in estimating the risk of, i.e. the target perimeter $P_T$. Furthermore, some refinements of the perimeter are required according to the features that are used in the second step.

**Firm model.** The first step of the model, or firm model, uses four different features. The first is the $P(d)$ predicted by an internal ISP model based on financial features of the firm such as the amount borrowed by the firm. This model represents the standard econometric approach to modeling the default probability of a firm. We compute the $P(d)$ for the firm and transform it into a log-odds score by applying a logit function (*Local*).

To this feature, we add information on the $P(d)$ coming from the rating model of the specific firm, treated in the same way (*Rating*). The rating model is the official, regulated model used by the bank to assess the $P(d)$ of a firm on a longer time horizon (one year), and uses features from the balance sheet of the firm. While the rating model is updated at a much slower pace than internal one (i.e., once per year), it still offers a background reference point for the $P(d)$ of the given firm.

We also choose to add to the model one of the most predictive features currently used in the internal model: the numbers of days of overdraft in the last three months (*Overdraft*). We found via experimentation that adding this feature to the firm model improves its predictive performance.

Finally, we introduce a historical feature that looks at the past of the given firm (*Hist*). This boolean indicator is 1 if the firm has been in default at any point in its past. Indeed, a firm which has been in default in the past is statistically more likely to get into a situation of distress again.

The firm model is a reduced-form model that can be expressed by the formula

$$Y = f(Local, Rating, Overdraft, Hist),$$

where $Y$ is the binary target variable representing whether a firm is in default in the next three months. We train this model by estimating it on data from a sliding window of the historical dataset.

We experiment with several possible learning algorithms for the firm model, both linear and non-linear. In particular, we test Logistic Regression (LR), Random Forests (RF) [59], Gradient Boosted Decision Trees (GBDT), and Kernel SVMs. The prediction performance of Kernel SVMs is barely better than baseline LR (average R@K of 0.46 compared to a baseline of 0.42). Conversely, both RF and GBDT achieve a R@K around 0.64, which is considerably better than the baseline LR. Of the two tree-based models, RF is the more easily applicable one, as it allows to use instance weights to rebalance the skew in the classes. GBDT could also be applied, but it requires creating bootstrap samples to rebalance the classes externally. Therefore, we choose to use Random Forests as the learning algorithm for the firm model (step 1).

The hyper-parameters for the model are optimized via prequential cross-validation [60]. We use 500 trees, and a minimum number of instances per leaf of 5 (pre-pruning). All the other hyper-parameters do not affect the accuracy of the model. Fig 6 shows the performance

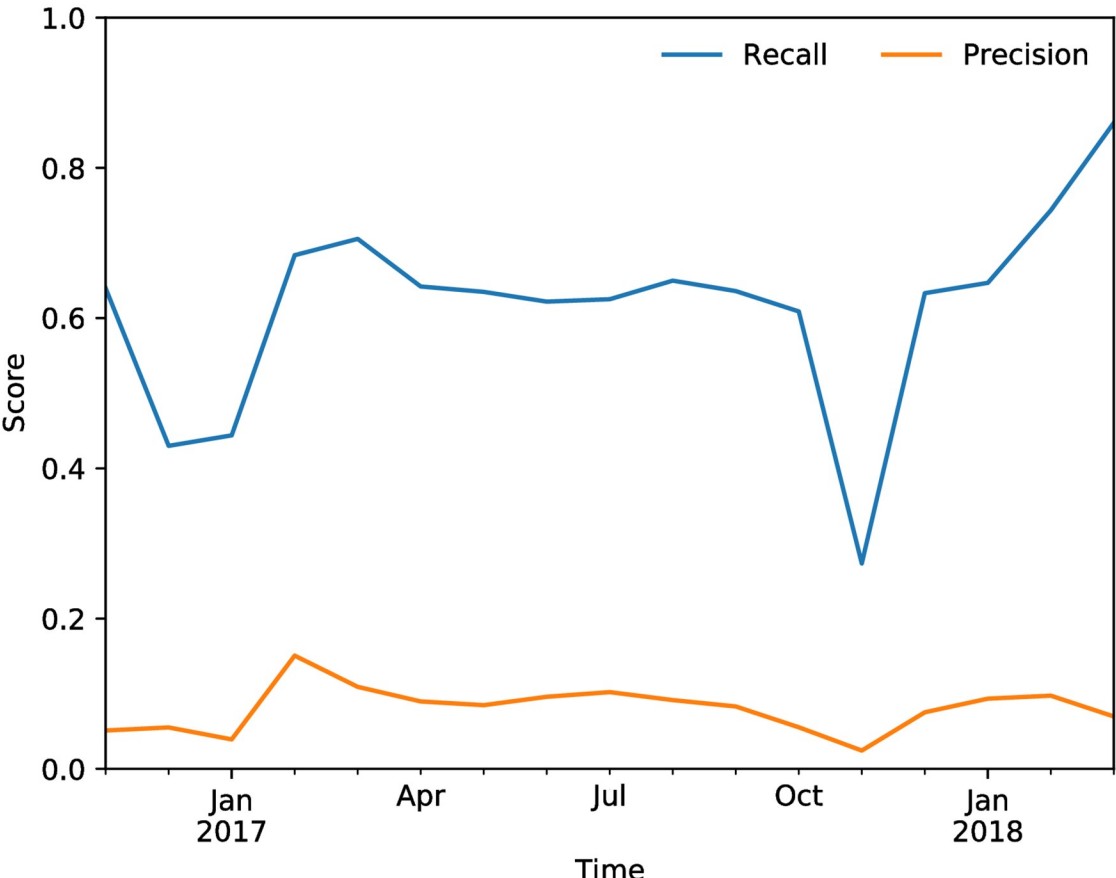

**Fig 6. Recall@K and Precision@K for the random forest model based on firm features as a function of time.** The average values for the score are R@K = 63.9%, P@K = 8.5%, F1 = 14.8%.

curves for the evaluation of the RF model. The model has a rather high average R@K of almost 64% for the period under consideration, which compares favourably with the average R@K of 53% of the linear regression model.

Fig 7 shows the average relative importance of the firm features as inferred by the Random Forest model over time. The scores from the two models (Rating and Local) are clearly the most important features, while the overdraft and historical default features have a more marginal contribution.

The output of the first step, the single-firm component, is a value for the $P(d)$ of each firm based conditioned on all the financial information available within the bank on the firm.

**Network spillover model.** The goal of the second network step is to provide an estimate of this $P(d)$ based on features coming from the network connections of the given firm. In other words, it estimates the risk imported from firm's customers and suppliers, by accounting for the firm fragility to shock imported from the supply chain. In particular, we make use of two main features: *fragility* and *personalized PageRank*. Note that the network model is estimated only on the target perimeter $P_T$.

*Fragility.* By granting trade credit, suppliers expose themselves to risk "'imported'" by their customers. The fragility feature captures the exposure of a given firm to risky customers in its 1-hop neighborhood (and symmetrically for suppliers), and it is a measure of the reliability of the trade credit of each firm.

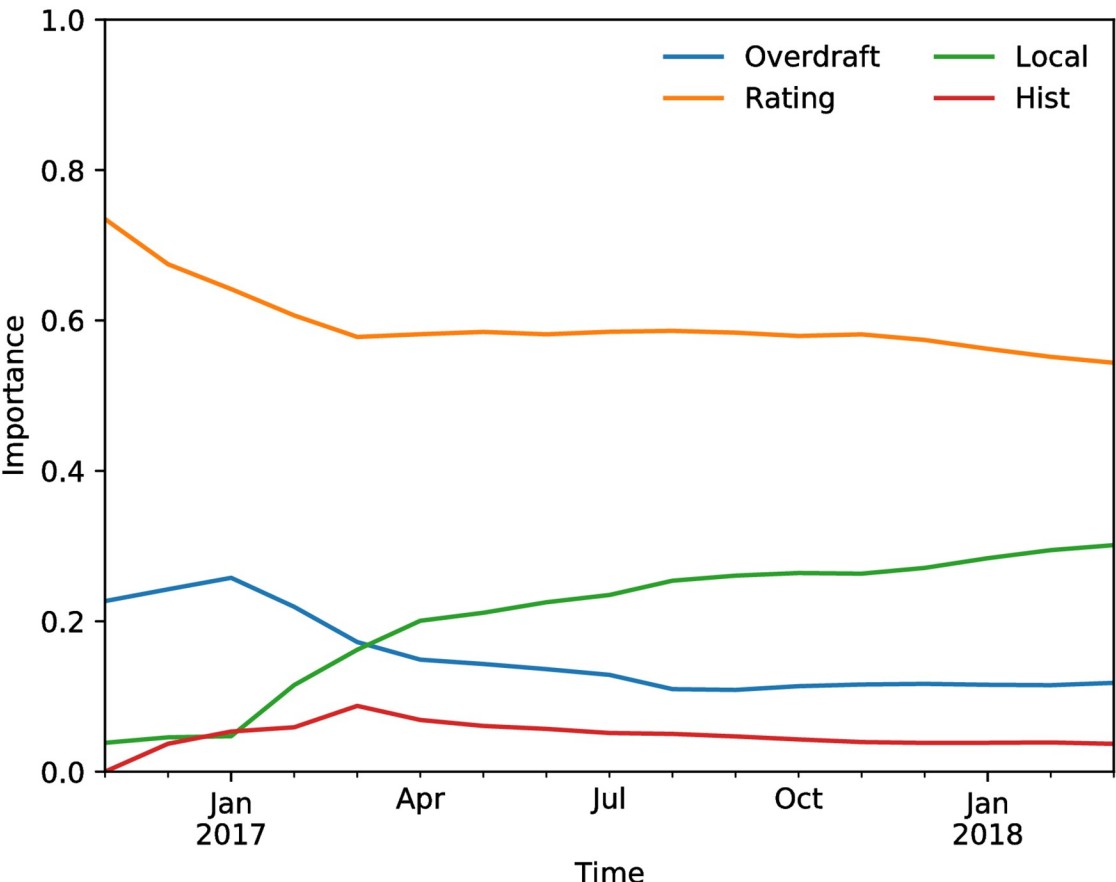

**Fig 7. Relative feature importance for the firm features in the random forest model.**

We assume that each firm is exposed to risk transmission via trade credit in proportion of which fraction of its revenue is represented by *account receivables*, i.e., claims for payment held by the firm for goods supplied or services rendered that customers have not paid for yet. This fraction of revenue is at risk if the customers of the given firm default or delay their payments.

The fact that suppliers grant credit, without often having the necessary structures to assess the underlying risk or the resources to manage any late payment, is considered one of the major exposures to failure of industrial companies [33]. Trade credit in the Italian market is a particularly significant form of financing due to both the structure of the production system, characterized by medium and small sized enterprises, and the operating practice applied in many sectors. In Italy this form of financing is often the result of habits, rather than a complement, or a substitute, for bank financing. Companies typically respond to late payments from their customers by in turn delaying payments to suppliers [22], thus generating a diffusion of the phenomenon through the network.

Thus, we model the dependency of the risk of the firm on the risk of the customers as a weighted average of the customers' $P(d)$, where the weights capture the relative strength of the connection between the firm and the customers.

An increase in trade credits $AR_i$ for the $i$ firm means greater exposure to the trade credit channel, hence a greater dependence on the firm's ability to service its debt on imported risk from customers. A firm with relatively higher $AR_i$, is more vulnerable to late payments from

customers and it is more likely to spread liquidity shocks to suppliers by delaying payments in turn. On the other hand, an increase in firm size, captured by higher net sales $S_i$, signals lower trade credit exposure. A larger size is also a sign of a lower likelihood of credit rationing: large firms have easier access to bank loans, so they are better able to absorb liquidity shocks. Customer payment delays are therefore less likely to spread to their suppliers. The expected sign of the $FRG_c$ coefficient value is positive.

A similar reasoning can be applied in the opposite direction, that is, by modeling how default risk spreads from suppliers to customers. In this case, the economic interpretation is more oriented to the market power of the customer within the supply chain. Larger customers, in terms of purchases, have greater market power, which is reflected in the ability to obtain deferred payments and other support measures from suppliers in the event of a liquidity shortage. Moreover, higher is the trade debt $AP_i$ of the customer $i$ owned by the supplier $j$, higher is the implicit stake of the supplier in its business. In other words, higher is the customer trade debt to its suppliers and higher is its sensitivity to the supplier's financial soundness. The $FRG_s$ coefficient value is also expected to be positive.

The final formulas for computing the fragility is specified as:

$$FRG_c(i) = \frac{AR_i}{S_i} \times \mathrm{logit}\left(\sum_{j \in \overleftarrow{N}(i)} w_{ji} P(d)_j\right), \tag{1}$$

$$FRG_s(i) = \frac{AP_i}{P_i} \times \mathrm{logit}\left(\sum_{j \in \overrightarrow{N}(i)} w_{ij} P(d)_j\right), \tag{2}$$

where $AR$ and $AP$ are account receivables and account payables, $S$ and $P$ are sales and purchases, $\overleftarrow{N}(i)$ and $\overrightarrow{N}(i)$ are the in-neighbors and out-neighbors of $i$ in the transaction network, $w_{ij}$ is the normalized weight of the edge between $i$ and $j$, and $P(d)_j$ is the probability of default of $j$ as computed by the model in the first step. We apply a logit transformation to the weighted average of the $P(d)$ of the neighbors in order to transform the probability into more suitable log-odds.

*Personalized PageRank.* The personalized PageRank (PPR) feature captures how close the given firm is in the network to the set of firms which have had a default event [61]. Firms with a default event can cause a liquidity shock that spreads through the network and cause distress to their supply chain. Therefore, we assume that default risk spreads through the transaction network as a random walker.

We also wish to model that firms closer to a defaulting customer are more likely to be affected, as seen in Section Basic definitions. Therefore, we impose a restart probability to the random walker: with probability $\alpha$ the random walker follows the transaction network, and with probability $1 - \alpha$ it restarts its random walk from its origin.

We also wish to model that being closer to multiple defaulting customers is likely to increase the risk of a firm. Therefore, we allow the random walk to (re)start from a set of *seed* nodes, represented by a distribution $Q$ over the nodes of the transaction network. For our application, $Q$ has non-zero values only for firms which have been in default (more on the choice of $Q$ later). $Q$ is also called a *restart vector*.

Finally, we compute the feature as the stationary distribution of the random walker described above. This distribution exists and is unique, and can be easily computed by the PPR algorithm. For every other node we compute the PPR from the restart vector as the solution to

the recurrent equation

$$PPR_\alpha = \alpha\, PPR_\alpha\, M + (1-\alpha)Q,$$

where $M$ is the row-stochastic adjacency matrix of the transaction network, $Q$ is the restart vector distribution, and $\alpha$ is a damping parameter $\in(0, 1)$. Finally, we normalize the $PPR_\alpha$ value obtained by the algorithm to reduce its bias towards high in-degree nodes

$$PPR(i) = \frac{PPR_\alpha(i)}{|\overleftarrow{N}(i)|},$$

where $|\overleftarrow{N}(i)|$ is the in-degree of node $i$. This normalization is also known as Effective Importance [62]. PPR is further normalized via standard score normalization to mean zero and unit standard deviation.

There are two free parameters in the algorithm to compute the PPR: the damping factor $\alpha$ and the restart vector $Q$. We optimize them by using prequential cross-validation on the training set. The best value for $\alpha = 0.25$ (as shown in Fig 8) indicates that the contagion does not spread very far from the initial seed nodes (which is consistent with the findings from the analysis of the default cascades, see the S1 Appendix). The results for $\alpha = 0.05$ are slightly better but we prefer to use a less-extreme value of the parameter as the difference is negligible.

For $Q$, we set a restart vector which at timestamp $t$ includes all the firms which have been in default at times $t' \leq t$. This set of seed nodes represent the ones that are contagious at time $t$. We try several temporal discounting factors (i.e., linear and exponential) to give more weight to more recent defaults. However, cross-validation results show that a simple uniform distribution over the vector performs best, as shown by Fig 9. We show the comparison between the results with $\alpha = 0.25$ and $\alpha = 0.85$ without temporal discounting and the results with linear and exponential discounting. We observe a decrease of the performance of the model with the linear temporal discounting while no effects with the exponential one. This result could indicate

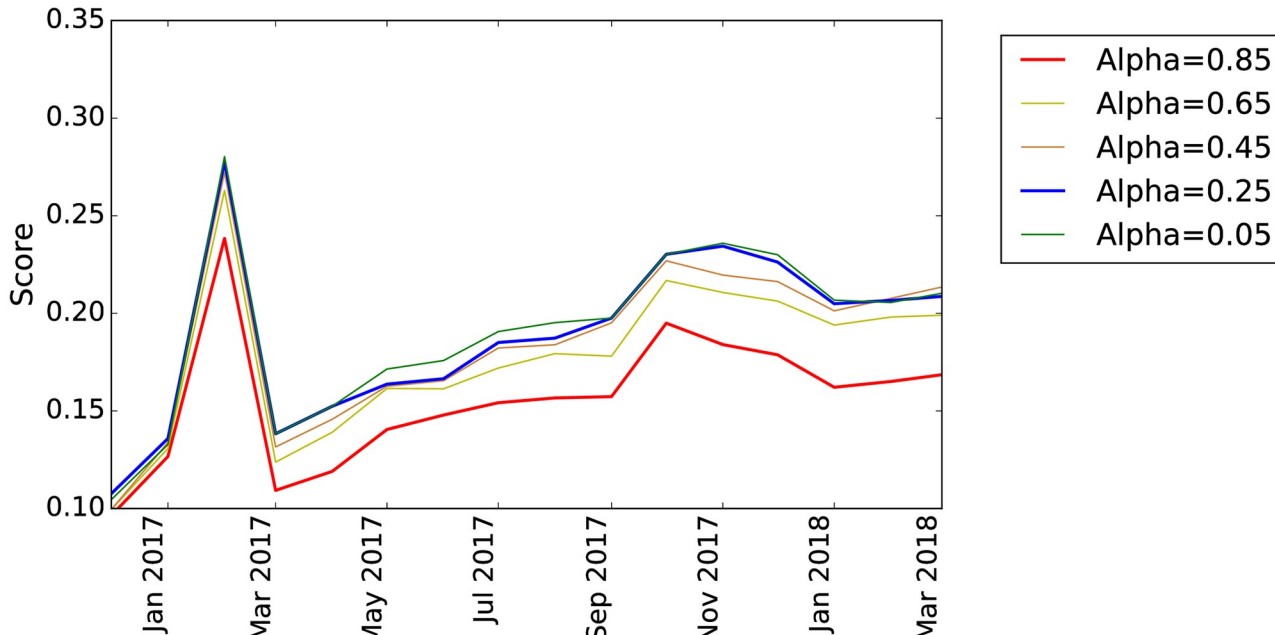

**Fig 8. Recall@5% for the network model using a single step model with the _PPR_ as unique feature for different values of _α_.**

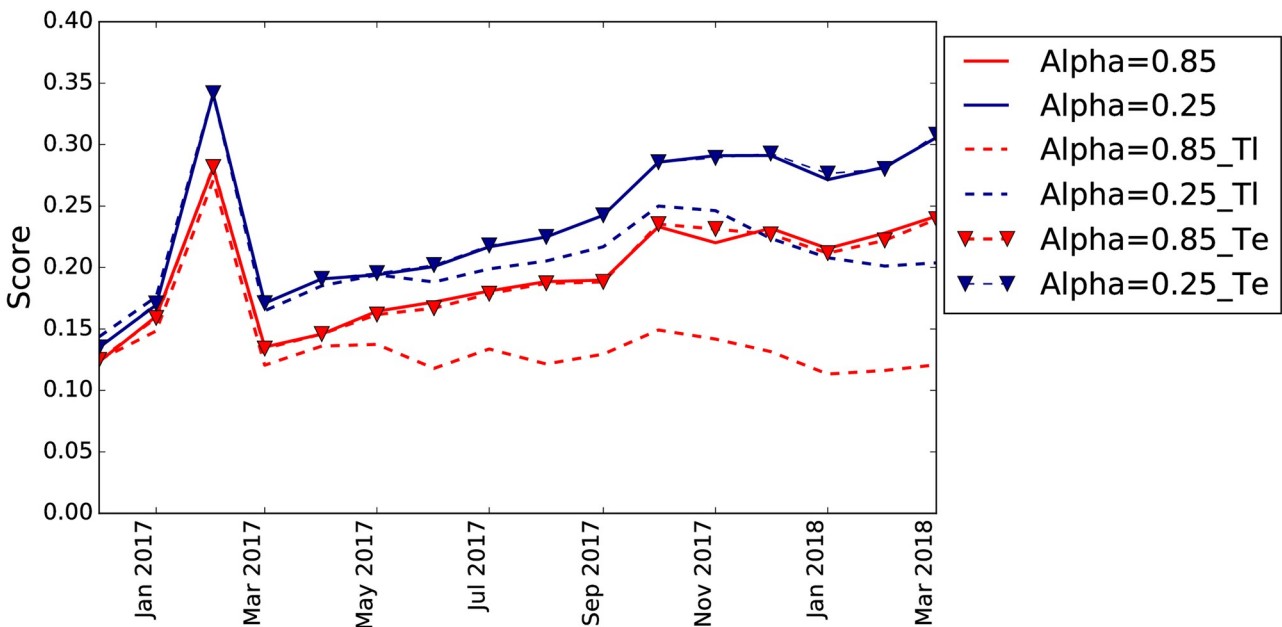

**Fig 9. Recall@5% for the network model using different temporal discounting for the restart vector Q.** The dashed lines correspond to the linear discount while the dashed lines with triangular markers to the exponential temporal discounting.

that the window of observation for the available data is too short to see any forgetting effect, in accordance to the results shown in the S1 Appendix.

*Network data coverage and instance weighting.* Our knowledge of the network is not complete and it is not homogeneous: on some companies we have an exhaustive view of transactions, while on others we have very partial information.

A partial view on a network subset could be the source of biases: firms with low number of visible links may be either conveying payments towards other banks, or belonging to a concentrated supply-chain, with very different implications on concentration risk and thus network spillover.

The more complete our knowledge of the supply-chain, the more reliable the estimate of the influence of the network on the risk of the company will be.

This is definitely the case for the fragility features, which explicitly rely on how much of the firm's financial position the network captures, but is also true indirectly for the PPR feature, as the presence or absence of a link (and its weight) clearly affects the random walks which the feature is based on.

For these reasons, we employ an instance weighting scheme, so that the model can focus on the data points which are more reliable. For each firm $i$, we define an instance weight for the machine learning model as:

$$W(i) = \frac{1}{2}\left(\frac{\sum_{j\in\overleftarrow{N}(i)} w_{ji}}{S_i} + \frac{\sum_{j\in\overrightarrow{N}(i)} w_{ij}}{P_i}\right).$$

The weight is therefore the average of the in-coverage and the out-coverage of the network with respect to the balance sheet data (sales $S$ and purchases $P$). More in detail, the first term is the ratio of the sum of in-weights of the network to the sales of the firm, while the second term is the ratio of the sum of out-weights of the network to the purchases of the firm. Therefore,

**Table 2. Example coefficients for one instance of the network model.**

| Feature | Coefficient | Std. Err. | z-value | p-value |
|---|---|---|---|---|
| $PPR$ | 2.97950 | 0.06246 | 47.695 | $<2 \cdot 10^{-16}$ *** |
| $FRG_c$ | 1.34992 | 0.16205 | 8.3299 | $<2 \cdot 10^{-16}$ *** |
| $FRG_s$ | 3.11574 | 0.13338 | 23.359 | $<2 \cdot 10^{-16}$ *** |
| $FRG_c \cdot FRG_s$ | -1.65613 | 0.46157 | -3.5880 | 0.0003 *** |

for a well-mapped firm this weight will be close to 1, while it will be close to 0 for firms which the network has little information on.

*Final network spillover model.* The overall combined model is as follows:

$$Y = f(PPR, FRG_c, FRG_s, FRG_c \cdot FRG_s),$$

Where *PPR* is the personalized PageRank, $FRG_c$ and $FRG_s$ are the two fragility terms for customer and supplier views, and $FRG_c \cdot FRG_s$ is an interaction feature for the two fragilities. As with the first step, we try several linear and non-linear classifiers. In this case, there is no significant difference in performance between LR and the other classifiers. Therefore we employ a linear regression model because of its superior interpretability.

Table 2 shows a summary of the model (trained on the whole timeline). All the features are statistically significant with $p \leq 0.001$. The signs of the coefficients for the PPR and Fragility (client and supplier) variables are positive, as expected. It is also reasonable to have a negative interaction term.

Moreover, the proportion among coefficients is an interesting result. In fact, here we can see that not only shocks can propagate in the direction of the flow of money through the network (coefficients of *PPR* and $FRG_c$) but also in the direction of the flow of goods and services exchanged (coefficient of $FRG_s$) and with a comparable intensity. As already mentioned, the latter interpretation is based on the idea that the default of a key-supplier can lead to the need for a costly search of a new supplier, with associated costs and possible production slowdowns.

Fig 10 shows the R@K and P@K for the target perimeter as a function of time. The network model captures a large fraction of the predictions of the single-firm model, without using information about the firm itself to make its prediction. Note that these results are meant to show the power of network information, and what is achievable without having direct informations about a firm. Clearly, the model by itself is not competitive with the one that uses single-firm features in terms of predictive performance. However, the numbers are not far off, which is a testimony to the importance of network features in this task. This importance is confirmed by the results of the hybrid model, shown next.

## Hybrid model

The hybrid approach merges the power of the network features from the network spillover approach described in Modeling fragility of firms to network spillovers, with single firm features, which are already known to be significant in this setting. The model captures the mix of effects by means of proper feature engineering and selection, by following a mostly data-driven approach. We detail the different phases of this approach next.

**Feature engineering.** We consider a wide array of features that could be potentially predictors for the default forecasting task, divided in the following high-level groups:

1. characteristics of the firm;

2. characteristics of the firm inside the network;

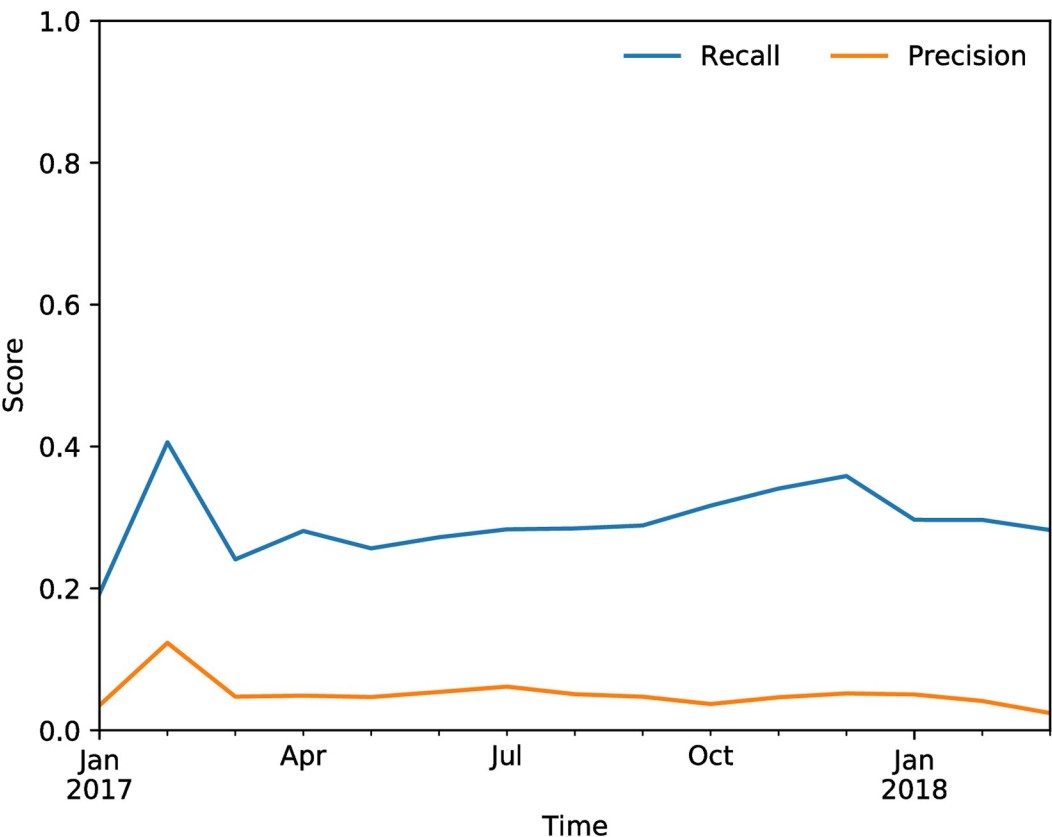

**Fig 10. Recall@K and Precision@K of the 2-step model on the target perimeter.**

3. characteristics that can spread through the network.

The first group includes single-firm features coming from different data sources, which present comprehensive picture of the firm from several angles (see Section Data description). These are firm features that are known to be predictive in this context from previous studies inside Intesa Sanpaolo, and some of these are used as linear covariates in a generalized linear model (GLM) to define the baseline.

The second set includes topological information about the firm in the network, constructed as explained in Section Network of Transactions. Some examples of features computed are the total number of neighbors, the total amount outgoing from the firm, the total amount incoming, and the ratio between outgoing and incoming amounts.

The third set aims at describing and summarizing the liquidity status of the firm's neighborhood. The most important firm features are summarized over all neighbors of each firm, by considering both the unweighted undirected version of the network and the weighted directed one. In the latter case, features are computed separately for suppliers and customers of the firm (out- and in-neighbors, respectively). We aggregate the neighborhood statistics by using their mean, but also their maximum, given the semantic of the features. Some examples include the maximum among neighbors' revenues and the weighted mean of account payables over purchases of the suppliers.

Finally, some trend features are computed as the difference between the value at month $t$ and the same value at month $t − 1$. These features may add information about the liquidity status of the neighbors getting worse than in the previous observation.

The result of this phase is a list of more than 300 possible features. Clearly, only some of them are really predictive, thus a feature selection is required.

**Feature selection.** The goal of this phase is to automatically select features that are predictive for the forecasting task. We first perform a screening on the usability of all the covariates to ensure they include no leaking of future information. Then, we proceed with the features selection on a subset of the dataset selected as follows:

- use only firms that are included in the target perimeter ($P_T$) and that were not already in default at the moment of prediction;

- only 4 time steps are included in the selection: 2017-03, 2017-06, 2017-09, 2017-12. These time steps are chosen such that the following 3 months would not overlap.

The global approach for feature selection follows two steps: univariate feature selection and feature collinearity.

*Univariate feature selection.* We use several tests statistics for each feature, depending on its nature (categorical or numerical):

1. chi-squared test, selecting features with p-value $\leq 0.01$;

2. t-test and variance test for numerical features, selecting features with p-value $\leq 0.01$ in both tests;

3. Kolmogorov-Smirnov test, selecting features with p-value $\leq 0.01$;

4. A set of model-based tests taking into account the average recall with a random forest, average recall with a GLM model with logit link, p-value of the coefficient estimated by the GLM model with logit link (in case of categorical variable, the minimum p-value). This set of tests aims at testing the predictive power of each single feature. Features are selected if they have all the following properties: an average in-sample recall greater than 0.7 with the random forest, an average in-sample recall greater than 0.5 with the GLM model, and the p-value of the coefficient $\leq 0.05$.

These filters reduce the number of features from 300 to less than 130 feature candidates.

*Feature collinearity.* Some features from the previous step could be redundant because the information they express is already contained in other features. To identify these features, we employ a model-based backward feature elimination step as follows:

1. Features are ordered from the least predictive to the most predictive one, according to the sum of the recall from GLM and random forest in the previous step.

2. For each feature $f$ in the previous ordered list, a predictive random forest model of $f$ is trained by using every other feature in the list.

3. If it is possible to predict $f$ with an $R^2$ greater than 0.8 or a prediction error smaller than 0.05 (in case of numerical or categorical feature respectively), then $f$ is explainable by the other features and it is filtered out from the feature set, otherwise it is kept.

This step produces a short list of 63 features. Among the most predictive ones there are both single firm and network ones (e.g., weighted mean of account payable over purchases of the clients considering the directed weighted network of payments). Finally, we perform a human validation step to substitute similar features for improved interpretability.

*Model training.* We test several models such as GLM [63], random forest [59], SVM [64], and XGBoost [65], and find the latter to be the best performing on a validation set. The hyperparameters are also optimized via a prequential cross-validation procedure.

**Table 3. Feature importance of the XGBoost model sorted by information gain.** Several network features populate the top of the list (the column Network indicates whether the feature is a network one).

| Feature | Gain | Cover | Network |
|---|---|---|---|
| Rating Class | 0.497 | 0.425 | |
| Std Days of Overdraft | 0.228 | 0.275 | |
| Max Days Overdraft 3months | 0.088 | 0.223 | |
| Rating Model | 0.058 | 0.042 | |
| Total amount incoming | 0.016 | 0.003 | Yes |
| black!30 Total amount outgoing | 0.013 | 0.002 | Yes |
| Foundation Date | 0.010 | 0.003 | |
| Revenues—Max over Neighbourhood | 0.009 | 0.001 | Yes |
| Max Days Overdraft 3months—Mean over Neighb | 0.008 | 0.002 | Yes |
| Purchases | 0.008 | 0.001 | |
| black!30 Uncommitted Credit in Italian Financial Institutions | 0.007 | 0.002 | |
| Short Term Debt over Revenues | 0.007 | 0.002 | |
| Region | 0.006 | 0.001 | |
| Uncommitted Debt—Difference Suppliers vs Customers | 0.006 | 0.002 | Yes |
| Committed Debt—Difference Suppliers vs Customers | 0.006 | 0.001 | Yes |
| black!30 Purchases in Services | 0.005 | 0.001 | |
| Committed Credit—Mean over Neighb | 0.005 | 0.001 | Yes |
| Short Term Debt over Revenues—Mean over Neighb | 0.004 | 0.000 | Yes |
| Account Payables over Purchase—Mean over Suppliers | 0.004 | 0.001 | Yes |
| Net Profit over Market Sector | 0.003 | 0.001 | |
| black!30 Account Payables over Purchase | 0.003 | 0.002 | |
| Account Receivables over Sales | 0.003 | 0.001 | |
| PPR | 0.003 | 0.001 | Yes |
| Short Term Debt over Sales—Max over Neighb | 0.002 | 0.001 | Yes |
| Flag for Past Default Events | 0.002 | 0.005 | |

Table 3 gives an interesting view of the feature importance as per the XGBoost model. Several network features populate the top of the list, e.g., 'Total amount incoming' (*Total amount towards the firm in payments*), and 'Revenues—Max over Neighborhood' (*Maximum of Revenues among the clients/suppliers considering the undirected unweighted network*). The importance of the network features demonstrates that the feature engineering and selection are effective in extracting and exploiting the additional signal with respect to the single firm features.

Fig 11 shows the R@K of the XGBoost model over time in comparison with two different baselines (both computed using prequential validation): a local GLM model which uses only single firm features, which represents the current state-of-the-art early warning system (EW Baseline), and a simplified version of the traditional Altman's credit risk model [44]. Given that most of the firms in our database are not publicly traded, we cannot use features such as market value of equity, thus we use a simplified version of the Altman's Z-Score. The features included are $X1$ = working capital / total assets, $X2$ = earnings before interest and taxes / total assets, and $X3$ = sales / total assets. Moreover, we re-estimated the coefficients on our training sample, and used our dependent variable $Y_i^t$, to make the performances comparable to the other models presented. We remark that Altman's model was developed for the US market, where companies are larger on average and have different financial characteristics from Italian companies, and it relies only on financial statements information, which are commonly

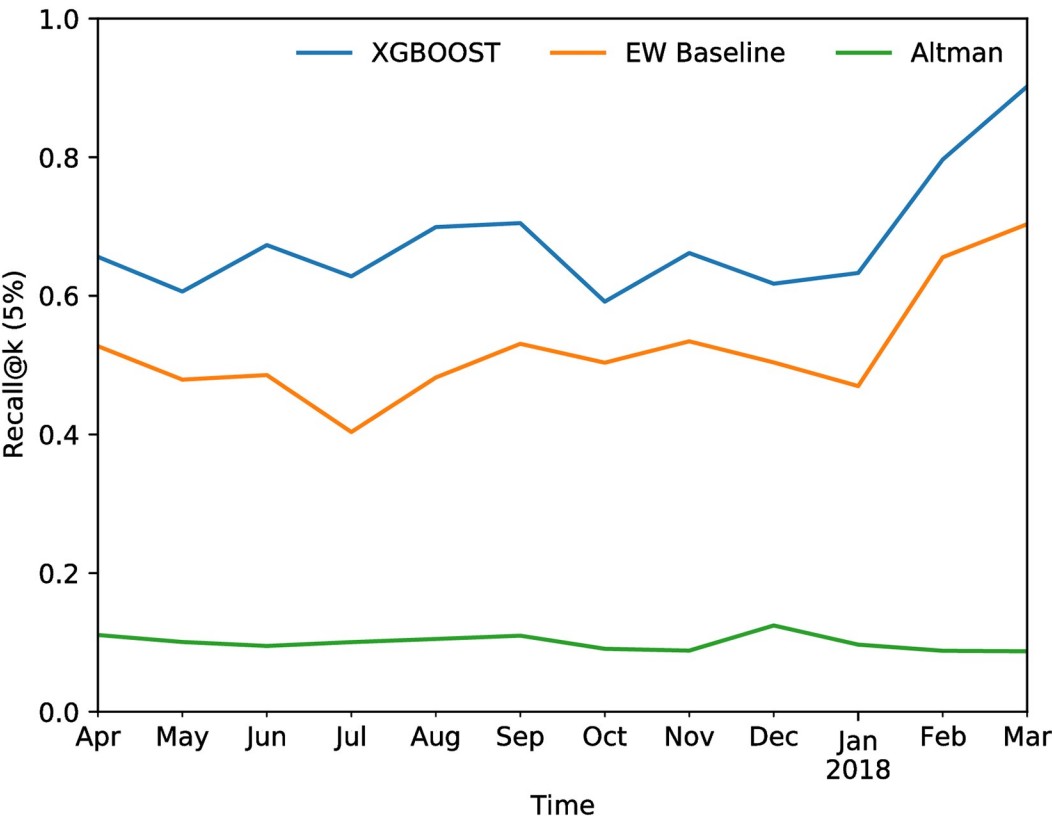

**Fig 11. Recall@K for the XGBoost model with mixed single-firm and network features as a function of time in the prequential setting compared to a logistic regression model on single-firm features (baseline).** The average R@K is 68.1% and the AUC is 90.5%.

updated yearly. Its poor performance is therefore expected on a short-term risk assessment task on Italian firms, as the one we tackle.

The average increase in performance of the hybrid model over the baseline early warning system exceeds 10 percentage points, which is remarkable given the challenging nature of the task.

## Deployment results

The two models under study (network spillover and hybrid) were additionally deployed in a pre-production environment of Intesa Sanpaolo for several months after the design phase concluded. To avoid overlapping the time windows of the target variable, we present three separate snapshots: 2018-12, 2019-03, and 2019-06.

We first look at the 2-step model results. Table 4 shows the AUC, R@K, and P@K for the target perimeter as a function of time. The average R@K is 29.2%, and the average AUC is 64.3%.

**Table 4. Prediction performance of the 2-step model on the target perimeter.**

| Month | AUC | Precision@K | Recall@K |
|---|---|---|---|
| 2018 12 | 69.2 | 3.1 | 32.4 |
| 2019 03 | 61.6 | 3.6 | 25.5 |
| 2019 06 | 63.0 | 4.5 | 29.7 |

**Table 5. Performance of the XGBoost model with respect to the baseline on 3 out-of-time snapshots.**

| Month | AUC | | P@K | | R@K | |
|---|---|---|---|---|---|---|
| | Baseline | XGBoost | Baseline | XGBoost | Baseline | XGBoost |
| 2018 12 | 68.0 | 91.4 | 3.9 | 6.5 | 40.8 | 68.4 |
| 2019 03 | 86.3 | 91.9 | 7.6 | 9.9 | 54.3 | 70.0 |
| 2019 06 | 85.6 | 89.8 | 7.2 | 9.6 | 47.6 | 63.6 |

For the hybrid model, Table 5 shows the performance measures in the out-of-time test set, which are stable and coherent with respect to the validation phase: the mean recall of the XGBoost is 67.3%, compared to 68.1% in the validation. This result confirms the model and the tuning of the hyper-parameters are not overfitted on the data available during the analysis phase. The hybrid XGBoost models shows a substantial improvement with respect to the baseline single-firm model, with a 19.8 percentage points average improvement on the Recall@K.

## Conclusions

We presented two network-based models for forecasting the default of firms towards a bank: a 2-step model which uses only network information, and a hybrid one that mixes single-firm features with network ones. The models looks at the transaction network of the firm, as inferred from the payments transiting via the bank, in order to identify the trade partners of the firm. By using several features extracted from the network, they are able to predict the default of firms three months in advance. Their performance compares reasonably well to a baseline model which uses uses only single-firm features. On an out-of-time test set, the 2-step model has an average Recall@5% of 29.2%, compared to a baseline of 47.5%, while the hybrid model reaches 67.3%, an improvement of almost 20 percentage points. We thus show that the introduction of network features that model the trade credit of firms has a substantial impact on the task of short-term forecasting of defaults of firms.

The work shown here constitutes a first approach on how to leverage trade credit, and its role as a vehicle for transmission of risk, to improve the accuracy of a model in the default forecasting task. The two main outstanding issues are (*i*) the coverage of the network, which is only partial, and gives us only a small window of information on the underlying process, and (*ii*) the time horizon of the contagion process, which is unknown and might be actually larger than the observation window. For the first issue, we used a network enrichment strategy via record linkage and an instance weighting scheme to cope with the partial and biased view offered by the network, however having more data to get a complete picture of the network would solve the issue. For the second issue, while having more data would help determine the time horizon of the contagion process, if the process itself is slow and has a large variance, then there may be an inherent limit to how useful trade credit information is in forecasting a 3-months-in-advance default event.

## Supporting information

**S1 Appendix.**
(PDF)

## Acknowledgments

The research was conducted under a cooperative agreement between Fondazione ISI, Intesa Sanpaolo Innovation Center, and Intesa Sanpaolo.

The authors would like to thank Ilaria Sangalli for her inspiring ideas. We wish to thank also Silvia Ronchiadin, Laura Li Puma, Luigi Ruggerone, Guido Genero, Silvio Cuneo, Davide Alfonsi, Paolo Di Biasi, Fiorella Salvucci, Annalisa Richetto, Mauro Di Pietro, Rachele Desiante, Daniele Amberti, Mario D'Almo, and Andrea Cosentini for their support and useful comments.

## Availability of data and materials

The data that support the findings of this study are available from Intesa Sanpaolo but restrictions apply to the availability of these data, which were used under license for the current study, and so are not publicly available. Data are however available from the authors upon request and with permission of Intesa Sanpaolo.

## Author Contributions

**Conceptualization:** Gianmarco De Francisci Morales, Greta Greco, Marco Lamieri.

**Data curation:** Claudia Berloco, Daniele Frassineti, Greta Greco, Hashani Kumarasinghe, Shuyi Yang.

**Formal analysis:** Gianmarco De Francisci Morales, Daniele Frassineti, Greta Greco, Hashani Kumarasinghe, Shuyi Yang.

**Methodology:** Gianmarco De Francisci Morales, Greta Greco, Marco Lamieri, Shuyi Yang.

**Project administration:** Claudia Berloco, Arianna Miola.

**Software:** Gianmarco De Francisci Morales, Daniele Frassineti, Greta Greco, Hashani Kumarasinghe, Emanuele Massaro, Shuyi Yang.

**Supervision:** Claudia Berloco, Gianmarco De Francisci Morales, Marco Lamieri.

**Validation:** Claudia Berloco, Daniele Frassineti, Shuyi Yang.

**Visualization:** Claudia Berloco, Daniele Frassineti, Emanuele Massaro.

**Writing – original draft:** Claudia Berloco, Gianmarco De Francisci Morales, Greta Greco, Marco Lamieri, Emanuele Massaro, Arianna Miola, Shuyi Yang.

**Writing – review & editing:** Claudia Berloco, Gianmarco De Francisci Morales, Greta Greco, Marco Lamieri, Emanuele Massaro, Arianna Miola, Shuyi Yang.

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
