## [Decision Letter · Decision Letter 0]

22 Dec 2020

PONE-D-20-32164

Predicting Corporate Credit Risk: Network Contagion via Trade Credit

PLOS ONE

Dear Dr. De Francisci Morales,

Thank you for submitting your manuscript to PLOS ONE. After careful consideration, we feel that it has merit but does not fully meet PLOS ONE’s publication criteria as it currently stands. Therefore, we invite you to submit a revised version of the manuscript that addresses the points raised during the review process.

We look forward to receiving your revised manuscript.

Kind regards,

Raffaella Calabrese

Academic Editor

PLOS ONE

Journal Requirements:

'The authors received no specific funding for this work.'

We note that one or more of the authors are employed by commercial companies: Intesa Sanpaolo, Novartis Farmaceutica, Zone24x7.

a. Please provide an amended Funding Statement declaring these commercial affiliations, as well as a statement regarding the Role of Funders in your study. If the funding organization did not play a role in the study design, data collection and analysis, decision to publish, or preparation of the manuscript and only provided financial support in the form of authors' salaries and/or research materials, please review your statements relating to the author contributions, and ensure you have specifically and accurately indicated the role(s) that these authors had in your study. You can update author roles in the Author Contributions section of the online submission form.

Within your Competing Interests Statement, please confirm that these commercial affiliations do not alter your adherence to all PLOS ONE policies on sharing data and materials by including the following statement: "This does not alter our adherence to  PLOS ONE policies on sharing data and materials.” (as detailed online in our guide for authors http://journals.plos.org/plosone/s/competing-interests) . If this adherence statement is not accurate and  there are restrictions on sharing of data and/or materials, please state these. Please note that we cannot proceed with consideration of your article until this information has been declared.

3. Please remove your figures from within your manuscript file, leaving only the individual TIFF/EPS image files, uploaded separately.  These will be automatically included in the reviewers’ PDF.

Reviewers' comments:

Reviewer's Responses to Questions

**Comments to the Author**

1. Is the manuscript technically sound, and do the data support the conclusions?

Reviewer #1: Yes

Reviewer #2: Yes

2. Has the statistical analysis been performed appropriately and rigorously? 

Reviewer #1: Yes

Reviewer #2: Yes

3. Have the authors made all data underlying the findings in their manuscript fully available?

Reviewer #1: No

Reviewer #2: No

4. Is the manuscript presented in an intelligible fashion and written in standard English?

Reviewer #1: Yes

Reviewer #2: Yes

5. Review Comments to the Author

Reviewer #1: This paper develops a model to predict firm default based on a supplier-customer transaction network, used as a proxy for the trade credit (i.e. the payment extension granted by the supplier firm to the customers) network. With that purpose, two models are shown: the first one includes two different but strictly related "sub-models", i.e. the standard econometric method of computing the probability of default, P(d), with the conditioning financial information owned by the bank and a more complex model which leverages the novelty of the network characteristics and integrates them in the predictive model, taking into account not just their own risk but also the risk deriving from the neighborhood of customers and suppliers. The second model presented combines the aforementioned approaches by creating a wide set of features potentially able to predict the firm default. At a later stage, these potentially predictive features are selected and corrected for collinearity. Several prediction models are compared and XGBoost appears to be the most efficient.

The paper shows a sound approach to future firm default prediction, it is well structured and highlights clearly its main contributions with respect to the current state of art. The references list is comprehensive. Yet, some issues need to be addressed or clarified:

Some notation, even if already correct, can be improved for the sake of clarity. Although later specified, AR and FGR are used before a proper definition. Moreover, "Larger customers, in terms of P_i" would be definitely clearer if 'P_i' was replaced by 'purchases', independently from its definition.

Plots could be made clearer. For instance, in figure 2b the colours chosen does not allow to distinguish between the two degree distributions of the transaction network. The following score plots present blurred labels. In figure 11 one of the time-tick labels is misplaced. Also, some of the plots in the 'supporting information' sheets some plots are blurred and then not clear.

I personally find slightly misleading the following quote from the abstract: "which can represent a vulnerability to customer decisions for suppliers". To avoid ambiguity, given the importance of the abstract, I suggest to rephrase.

From the abstract it is mentioned that two models are presented, but from my understanding the first one (especially the second step) is used to test network features (or in general, features) can be efficiently integrated in the following hybrid model. My concerns (to be clarified and not necessarily to cause any changes in the paper) regard the method itself which could be seen as constituted by two blocks: first, the feature creation and valuation and second, the hybrid model for the forecasts.

As for the human validation step for the hybrid model, I can understand what could have guided you in the substituting process (common sense, in all likelihood), but it is not clear what could have guided you in the removing process. I would be grateful if you could further clarify.

Both the R@K and P@K shows negative peaks for the 2017Q4 for the Random Forest model (two steps model, step 1) and for the hybrid model. Could it be affected by any systematic feature in the dataset? Have you investigated further?

Reviewer #2: This paper is written in good english and it is well structured. The study is interesting and the authors use a unique data-set. However, this paper has several shortcomings. 1) I feel that the text is often vague, especially in the section "Data Description". For example, it is not clear which information from yearly financial statements is used and what the official risk measures computed by the bank are. 2) In many parts of the papers relevant citations are missing. For example, on page 4 the authors write "aligned with typical early warning models", but they do not explain what typical early warning models are nor do they cite relevant literature. 3) The authors do not compare their results with state of the art credit risk models. 4) The figures are often pixelated and not displayed in a consistent form throughout the paper. 5) Often, it is not clear why the authors perform certain analyses and not others.

6. PLOS authors have the option to publish the peer review history of their article (what does this mean?). If published, this will include your full peer review and any attached files.

Reviewer #1: No

Reviewer #2: No

---

## [Author Response · Author response to Decision Letter 0]

15 Mar 2021

We thank the editor for handling our manuscript, we welcome the comments of the reviewers, and are grateful for the opportunity to revise and improve our manuscript.

Here a point-by-point list of answers to the questions raised in the reviews, and changes made to the paper to address them.

To the Editor:

1. We have added a funding statement that identifies the industrial affiliations of some of the authors and the role of the funder to the cover letter.

2. We have added our competing interests statement, which already listed the industrial affiliations of some of the authors and declares that “This does not alter our adherence to PLOS ONE policies on sharing data and materials”, to the cover letter.

3. We have removed the figures from the main file.

4. The supporting information, which is an appendix section, already contains captions for all the figures contained in it. We have separated the appendix in the supporting information, and added a section reference to it at the bottom of the main file.

To Reviewer 1:

1. Fixed typos and poorly worded sentences.

2. Clarified the first sentence of the abstract.

3. Re-plotted Figures 2, 3, 6, 7, 11, S1, S7, S8.

4. Clarified that we present the hybrid model, and the network model is propaedeutic to the final one. Changed the intro to reflect this fact.

5. Clarified that we do not remove additional features in the human validation, it was poorly worded, we simply substitute features for better interpretation. Fixed the text.

6. We checked the performance drop of the RF model in 2017Q4 (November), but we did not find any systematic feature of the dataset that is clearly responsible for the drop (in terms of feature/class distribution). The hybrid model, however, does not seem affected by it, and the performance is in line with the other periods.

To Reviewer 2:

1. Added a section explaining the data we use from the various sources and which risk measures the bank already computes, according to the Basel framework.

2. Added a section on Model-based approaches that explains the most used models for bankruptcy modeling and credit risk, which explains what an Early Warning System is, and added the relevant citations.

3. Added a benchmark of a standard model for credit risk (Altman’s model) and compared to it in Figure 11. 

4. Re-plotted figures in vectorial format to avoid blurriness.

5. Improved the flow of the paper, we hope the message is clearer now.

We have also polished the text and improved the flow of the paper. We hope that the revised and improved paper addresses the main concerns of the reviewers.

---

## [Decision Letter · Decision Letter 1]

31 Mar 2021

Predicting Corporate Credit Risk: Network Contagion via Trade Credit

PONE-D-20-32164R1

Dear Dr. De Francisci Morales,

We’re pleased to inform you that your manuscript has been judged scientifically suitable for publication and will be formally accepted for publication once it meets all outstanding technical requirements.

Kind regards,

Stefan Cristian Gherghina, PhD. Habil.

Academic Editor

PLOS ONE

Additional Editor Comments (optional):

Reviewers' comments:

Reviewer's Responses to Questions

**Comments to the Author**

1. If the authors have adequately addressed your comments raised in a previous round of review and you feel that this manuscript is now acceptable for publication, you may indicate that here to bypass the “Comments to the Author” section, enter your conflict of interest statement in the “Confidential to Editor” section, and submit your "Accept" recommendation.

Reviewer #1: All comments have been addressed

Reviewer #2: All comments have been addressed

2. Is the manuscript technically sound, and do the data support the conclusions?

Reviewer #1: Yes

Reviewer #2: Yes

3. Has the statistical analysis been performed appropriately and rigorously? 

Reviewer #1: Yes

Reviewer #2: Yes

4. Have the authors made all data underlying the findings in their manuscript fully available?

Reviewer #1: Yes

Reviewer #2: No

5. Is the manuscript presented in an intelligible fashion and written in standard English?

Reviewer #1: Yes

Reviewer #2: Yes

6. Review Comments to the Author

Reviewer #1: All the comments done by me have been addressed. The sentences have been clarified accordingly and the figures are now definitely clearer. The authors have also been addressed and commented properly my remark on the Random Forest model.

Reviewer #2: (No Response)

7. PLOS authors have the option to publish the peer review history of their article (what does this mean?). If published, this will include your full peer review and any attached files.

Reviewer #1: No

Reviewer #2: No

---

## [Editor Report · Acceptance letter]

6 Apr 2021

PONE-D-20-32164R1 

Predicting Corporate Credit Risk: Network Contagion via Trade Credit 

Dear Dr. De Francisci Morales:

I'm pleased to inform you that your manuscript has been deemed suitable for publication in PLOS ONE. Congratulations! Your manuscript is now with our production department. 

Kind regards, 

on behalf of

Dr. Stefan Cristian Gherghina 

Academic Editor

PLOS ONE